# A global resource for genomic predictions of antimicrobial resistance and surveillance of *Salmonella* Typhi at pathogenwatch

Silvia Argimón [1✉], Corin A. Yeats[2], Richard J. Goater [1,10], Khalil Abudahab[1], Benjamin Taylor [2], Anthony Underwood[1], Leonor Sánchez-Busó [2,11], Vanessa K. Wong[3], Zoe A. Dyson [3,4,5], Satheesh Nair [6], Se Eun Park [7], Florian Marks [7], Andrew J. Page[8,12], Jacqueline A. Keane[8], Stephen Baker[9], Kathryn E. Holt[4,5], Gordon Dougan[3] & David M. Aanensen[1,2✉]

As whole-genome sequencing capacity becomes increasingly decentralized, there is a growing opportunity for collaboration and the sharing of surveillance data within and between countries to inform typhoid control policies. This vision requires free, community-driven tools that facilitate access to genomic data for public health on a global scale. Here we present the Pathogenwatch scheme for *Salmonella enterica* serovar Typhi (*S.* Typhi), a web application enabling the rapid identification of genomic markers of antimicrobial resistance (AMR) and contextualization with public genomic data. We show that the clustering of *S.* Typhi genomes in Pathogenwatch is comparable to established bioinformatics methods, and that genomic predictions of AMR are highly concordant with phenotypic susceptibility data. We demonstrate the public health utility of Pathogenwatch with examples selected from >4,300 public genomes available in the application. Pathogenwatch provides an intuitive entry point to monitor of the emergence and spread of *S.* Typhi high risk clones.

[1] Centre for Genomic Pathogen Surveillance, Wellcome Genome Campus, Hinxton, Cambridgeshire, UK. [2] Centre for Genomic Pathogen Surveillance, Big Data Institute, Nuffield Department of Medicine, University of Oxford, Oxford, Oxfordshire, UK. [3] Addenbrooke's Hospital, Cambridge University Hospitals NHS Foundation Trust, Cambridge Biomedical Campus, Cambridge, UK. [4] London School of Hygiene and Tropical Medicine, London, UK. [5] Department of Infectious Diseases, Monash University, Melbourne, Australia. [6] Gastrointestinal Bacterial Reference Unit, Public Health England, Colindale, London, UK. [7] International Vaccine Institute, Seoul, Republic of Korea. [8] Pathogen Informatics, Wellcome Sanger Institute, Wellcome Genome Campus, Hinxton, Cambridgeshire, UK. [9] Cambridge Institute of Therapeutic Immunology & Infectious Disease, Department of Medicine, University of Cambridge, Cambridge, UK. [10] Present address: Wellcome Sanger Institute, Wellcome Genome Campus, Hinxton, Cambridgeshire, UK. [11] Present address: Genomics and Health Area, Foundation for the Promotion of Health and Biomedical Research in the Valencian Community (FISABIO-Public Health), Valencia, Spain. [12] Present address: Quadram Institute Bioscience, Norwich Research Park, Norwich, Norfolk, UK. ✉email: silvia.argimon@cgps.group; david.aanensen@cgps.group

The ability to rapidly sequence microbial genomes facilitates the tracking of pathogen evolution in real-time and with a geographical context. Genomic surveillance provides the opportunity to identify the emergence of genetic signatures indicating antimicrobial resistance (AMR), or host adaptation, facilitating early intervention and minimizing wider dissemination. Consequently, genomic data has the ability to transform the way in which, we manage the emergence of microbes that pose a direct threat to human health in real time.

Genomic data is being generated at a remarkable rate, but we need to bridge the gap between genome science and public health with tools that make these data broadly and rapidly accessible to those who are not expert in genomics. To maximize the impact of ongoing surveillance programs, these tools need to quickly highlight high-risk clones by assigning isolates to distinct lineages and identifying genetic elements associated with clinically relevant features such as AMR or virulence. In this way, new isolates can be examined against the backdrop of a population framework that is continuously updated, and that enables both the contextualization of local outbreaks and the interpretation of global patterns.

*Salmonella* Typhi (*S.* Typhi) causes typhoid (enteric) fever, a disease that affects approximately 20–30 million people every year[1,2]. The disease is predominant in low-income communities, where public health infrastructure is poorly resourced. Similar to other infections, typhoid treatment is compromised by the emergence of *S.* Typhi with resistance to multiple antimicrobials, including those currently used for treatment[2]. Whole genome sequencing (WGS) has proven key to identify *S.* Typhi high-risk clones by linking the population structure to the presence of AMR elements. For example, the resurgence of multidrug resistant (MDR) typhoid (defined as resistance to chloramphenicol, ampicillin, and co-trimoxazole) has been explained in part by the global spread of an MDR *S.* Typhi lineage known as haplotype H58 or subclade 4.3.1[3,4], which is associated with both acquired AMR genes and fluoroquinolone resistance mutations[3,5].

WGS is increasingly being implemented in local and national public health laboratories, and web applications can provide rapid analysis and access to actionable information for infection control in the context of a global population framework. Online resources are available for the identification of acquired AMR mechanisms in bacterial pathogens, including *Salmonella spp.*[6,7], and for in silico typing and visualization of genome variation and relatedness based on WGS data[8–12]. Here, we describe Typhi Pathogenwatch, a web application to support genomic epidemiology and public health surveillance of *S.* Typhi. Typhi Pathogenwatch rapidly places new genomes within the broader geographic and population context, predicts their genotype according to established nomenclatures[4,8,13], and detects the presence of AMR determinants and plasmid replicon genes to assess public health risk. Results can be downloaded or shared via a web address containing a unique collection identifier. Our approach allows the rapid incremental addition of new data and can be used to underpin the international surveillance of typhoid, MDR, and other public health threats.

## Results

**Overview of Typhi pathogenwatch**. We developed a public health focused application for *S.* Typhi genomics that uses genome assemblies to perform three essential tasks for surveillance and epidemiological investigations, i.e., (i) placing isolates into lineages or clonal groups, (ii) identifying their closest relatives and linking to their geographic distribution, and (iii) detecting the presence of genes and mutations associated with AMR. The application can be accessed at https://pathogen.watch/styphi,

where users can create an account to upload and analyse their genomes (Fig. 1 and video[14]). User data remains private and stored in their personal account. Pathogenwatch provides compatibility with typing information for MLST[13], cgMLST[8], in silico serotyping (SISTR[11]), a SNP genotyping scheme (GenoTyphi[4]), and plasmid replicon sequences[15]. The results for a single genome are displayed in a genome report that can be downloaded as a PDF. The results for a collection of genomes can be viewed online and downloaded as trees and tables of genotypes, AMR predictions, assembly metrics, and genetic variation. Results can also be accessed at a later date and shared via a collection ID embedded in a unique weblink, thus facilitating collaborative surveillance.

**Clustering genomes into lineages with Pathogenwatch**. The pairwise genetic distance between isolates provides an operational unit for genomic surveillance. Typhi Pathogenwatch clusters genomes based on their genetic distance and displays their relationships in a collection tree. We benchmarked the Pathogenwatch clustering method against established methods of SNP-based tree inference, using three sets of published genomes. The Pathogenwatch trees clustered diverse genomes according to genotype assignments[4] (Supplementary Fig. 1a), and detected phylogeographic signal in a set of closely related genomes from a clonal expansion of 4.3.1 within Africa[3] (Supplementary Fig. 1b). In addition, we found that the Typhi Pathogenwatch clustering algorithm produced trees comparable to established methods based on the tree space (visualizations of pairwise distances between trees in two or three dimensions) and the tree topology (Supplementary Fig. 2).

**Contextualization with public data**. A fundamental process for interpreting genomic datasets is to identify the nearest neighbors to the genome(s) under investigation. Pathogenwatch contextualizes the user-uploaded genomes with public genomes using a population tree of 19 diverse genome references (Supplementary Fig. 3) to guide their SNP-based clustering into subsets of closely related genomes (population subtrees). A previous investigation of a typhoid outbreak in Zambia identified clonal diversity and two repertoires of AMR genes within outbreak organisms, which belonged to haplotype H58 (genotype 4.3.1)[16]. Using Pathogenwatch, the outbreak strains can be rapidly contextualized with public genomes, which revealed two different clusters with close relationships to contemporary genomes from neighboring countries Malawi and Tanzania (Fig. 2).

Users interested in exploring the public genomes without creating their own collections can browse the public data as a whole[17] or view by published study[18]. As of November 2020, Typhi Pathogenwatch included 4389 public genomes from 26 published articles (Supplementary Table 1). The genomes spanned the years 1905–2019 and 77 different countries, with the largest representation from 2000 onwards ($n = 3795$, 86.49%) and from the Indian subcontinent ($n = 1602$, 36.50%), respectively (Table 1 and Supplementary Fig. 4). Over half of the genomes ($n = 2500$, 57.0%) belonged to the globally dominant MDR genotype 4.3.1, although the five different genotypes comprising 4.3.1 showed different temporal distributions and relative abundance (Supplementary Fig. 5).

**Genotypic predictions of antimicrobial resistance**. Typhi Pathogenwatch queries genome assemblies with BLAST[19] and a curated library of AMR genes and mutations (Supplementary Table 2). The antibiotics table reports the presence of known AMR determinants as resistance, only discriminating between resistance and decreased susceptibility (intermediate) for ciprofloxacin. To benchmark the Typhi Pathogenwatch predictions, we

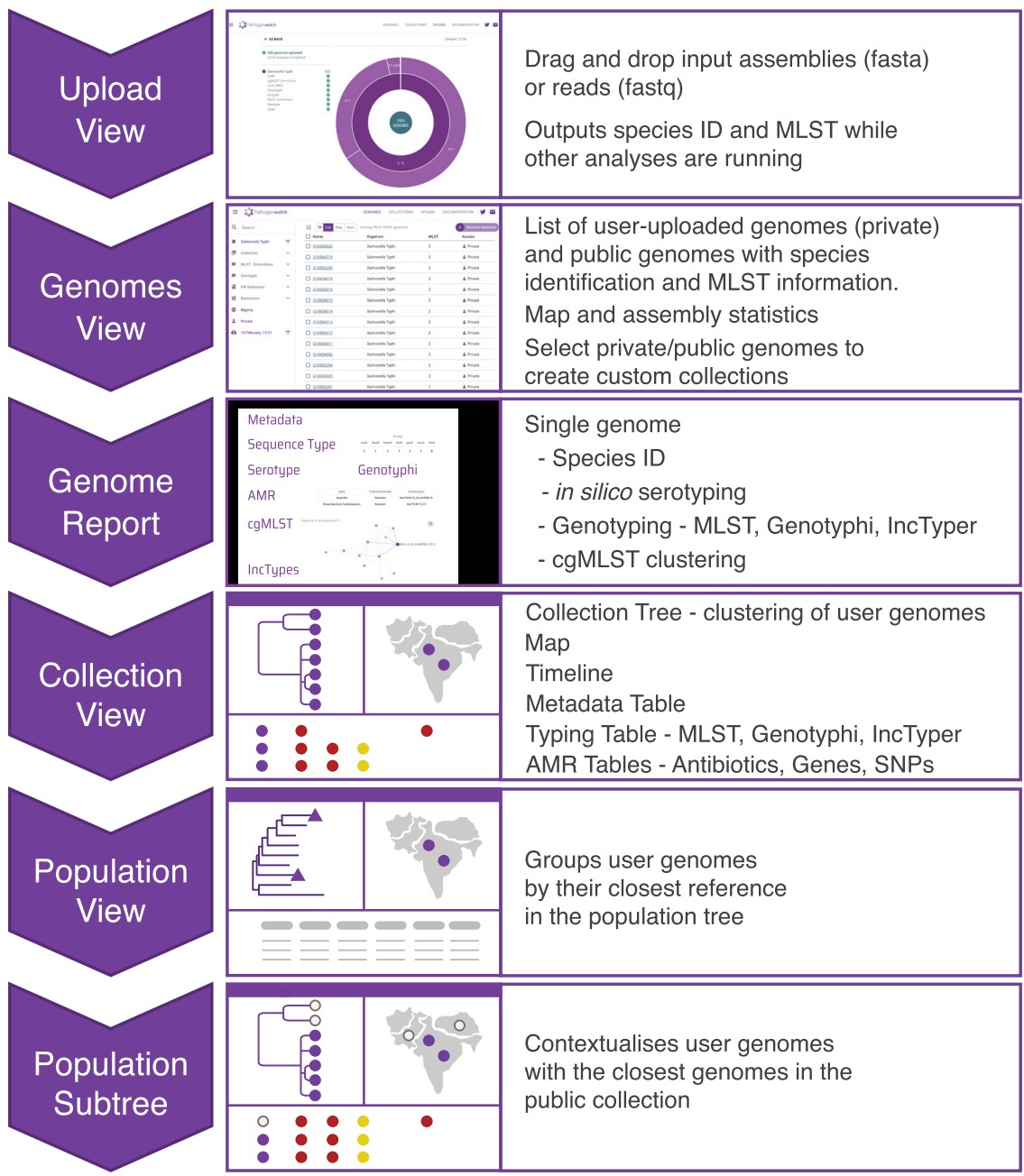

**Fig. 1 Workflow of the Typhi Pathogenwatch application.** Input assemblies or sequence reads and metadata files can be uploaded via drag-and-drop onto the Upload page. Once the analyses completed, the genomes are listed on the Genomes page with Pathogenwatch outputs for speciation and MLST. Clicking on a genome name on the list pops up a Genome Report. The user can create collections of genomes. The Collection view displays the user genomes clustered by genetic similarity on a tree, their location on a map, a timeline, as well as tables for metadata, typing and AMR. The Population view displays the user genomes by their closest reference genome in the population tree. Clicking on one of the highlighted nodes (purple triangles) opens the Population subtree view, which contextualizes the user genomes with the closest public genomes.

first compared the genotypic resistance genotypes to the available drug susceptibility phenotypes (SIR interpretation) of 1316 genomes. The sensitivity of the Pathogenwatch genotypic predictions was at least 0.96 for all antibiotics with a computed value (Table 2). The false negative (FN) calls for ampicillin ($n = 4$), cephalosporins ($n = 2$), chloramphenicol ($n = 6$), and sulfamethoxazole-trimethoprim ($n = 7$) were paralleled by the original genome studies[20–22], and by an alternative bioinformatics method[23]. The 49 FN calls for ciprofloxacin were also in agreement with the in silico analyses reported in the original genome studies[22,24], in which no QRDR mutations or *qnr* genes were

detected. Only mutations outside of the QRDR of *parE* (A364V, $n = 17$) or *gyrA* (D538N, $n = 2$) were found in 20 genomes.

The specificity of the Pathogenwatch genotypic predictions was at least 0.95 for most antimicrobials (Table 2), with the exception of ciprofloxacin, for which a third of the ciprofloxacin susceptible isolates were reported as insusceptible by Pathogenwatch. A closer inspection of the 57 false positive (FP) results showed that Pathogenwatch reported one ($n = 55$), two ($n = 2$), or three ($n = 1$) mutations in the QRDR of *gyrA, gyrB,* and/or *parC*, most frequently the single mutations *gyrA*_S83F ($n = 25$) and *gyrB*_S464F ($n = 16$). For 54 of these samples, the same

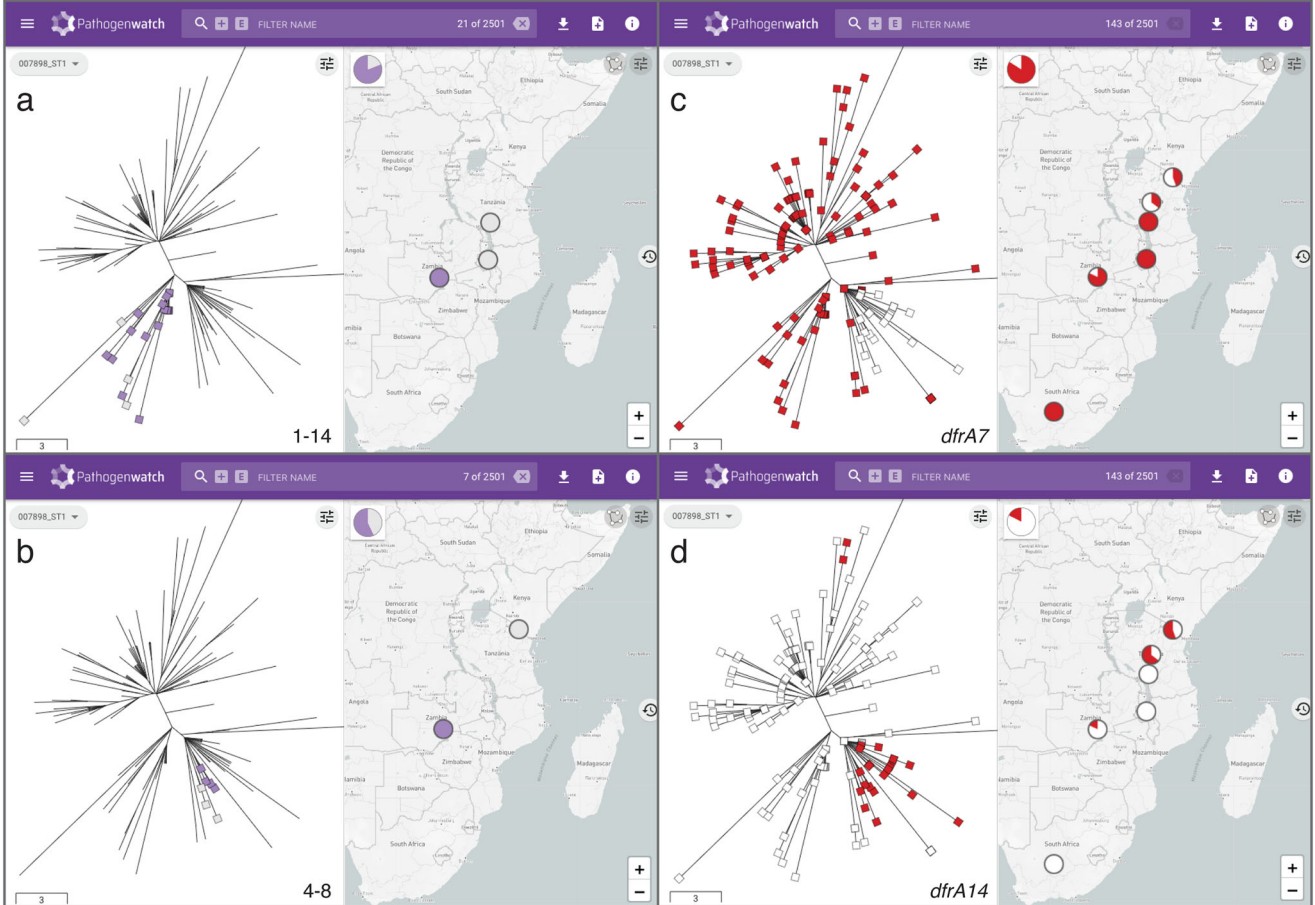

**Fig. 2 Pathogenwatch provides genomic context for outbreak investigations. a, b** Genomes from an outbreak in Zambia (purple markers on tree and map) are linked by genetic relatedness to genomes from neighboring countries Malawi and Tanzania (gray markers) forming two separate groups containing 16 (**a**) and 4 (**b**) outbreak genomes, respectively. The number of pairwise differences (range) between outbreak and related genomes in the Pathogenwatch score matrix are indicated on the bottom-right of the tree panel. **c, d** Differential distribution of trimethoprim resistance genes *dfrA7* (**c**) and *dfrA14* (**d**) across the two clades containing outbreak genomes. The presence of the *dfr* genes is indicated in red on the tree and map. The data are available at https://pathogen.watch/collection/g5pbucot6e58-hendriksen-et-al-2015.

### Table 1 Characteristics of 4389 public genomes in Pathogenwatch.

| Year of isolation | Number of genomes (%) |
|---|---|
| 1905–1969 | 41 (0.9) |
| 1970–1989 | 79 (1.8) |
| 1990–1999 | 395 (9.0) |
| 2000–2009 | 1187 (27.0) |
| 2010–2019 | 2609 (59.4) |
| No date | 78 (1.78) |

| Country of isolation (top 6) | Number of genomes (%) |
|---|---|
| Bangladesh | 637 (14.51) |
| United Kingdom | 629 (14.33) |
| India | 486 (11.07) |
| Nepal | 318 (7.25) |
| Vietnam | 220 (5.01) |
| Cambodia | 209 (4.76) |

| Assembly Stats | Median (range) |
|---|---|
| Number of contigs | 51 (1–633) |
| Assembly length | 4,747,975 (4,535,494–5,211,763) |
| N50 | 204,317 (19,527–4,806,333) |
| Non-ATCG | 152 (0–48,002) |
| GC content (%) | 52.0 (51.4–52.4) |

mutations were reported in the original genome studies. For the remaining three genomes, no mutations were reported in the original studies, but we confirmed the presence of *gyrB*_S464F ($n = 2$) or *gyrB*_S464Y ($n = 1$) in the assemblies using Resfinder[25].

To benchmark the predictions of ciprofloxacin resistance/decreased susceptibility, we then evaluated the ciprofloxacin MICs of 889 *S.* Typhi isolates from nine previous studies against the different combinations of resistance mechanisms identified by Pathogenwatch. The isolates with one or two QRDR mutations displayed mostly intermediate MICs against ciprofloxacin, and support reporting as intermediate in Pathogenwatch (Fig. 3). The MIC values of seven isolates carrying single mutations on *gyrA* (S83F, S83Y) and *gyrB* (S464F), however, were below the intermediate breakpoint, consistent with the lower specificity reported for ciprofloxacin in Table 2. The highest ciprofloxacin MIC values were observed for the combination of *gyrA*_S83F-*gyrA*_D87N-*parC*_S80I mutations, reported as resistant by Pathogenwatch[26–28]. However, the triple combination *gyrA*_S83F-*gyrA*_D87G-*parC*_E84K was represented by nine isolates with MICs in both the resistant ($n = 6$) and the intermediate ($n = 3$) ranges, and is reported by Pathogenwatch as intermediate. Further susceptibility testing of isolates with this combination of mutations is needed to refine genotypic predictions. Likewise, several other mechanisms potentially conferring insusceptibility

**Table 2 Benchmark of Typhi Pathogenwatch AMR predictions.**

| Antibiotic | Total | TN | TP | FN | FP | Sensitivity (95% CI) | Specificity (95% CI) | PPV (95% CI) | NPV (95% CI) | VME rate | ME rate | Concordance (%) |
|---|---|---|---|---|---|---|---|---|---|---|---|---|
| AMP | 875 | 461 | 402 | 4 | 8 | 0.99 (0.97–1.00) | 0.98 (0.97–0.99) | 0.98 (0.96–0.99) | 0.99 (0.98–1) | 0.01 | 0.02 | 98.63 |
| CEP | 348 | 256 | 90 | 2 | 0 | 0.98 (0.92–1.00) | 1.00 (0.99–1.00) | 1.00 (0.96–1.00) | 0.99 (0.97–1.00) | 0.02 | 0 | 99.43 |
| CHL | 913 | 518 | 375 | 6 | 14 | 0.98 (0.97–0.99) | 0.97 (0.96–0.99) | 0.96 (0.94–0.98) | 0.99 (0.98–1.00) | 0.02 | 0.03 | 97.81 |
| CIP | 1282 | 111 | 1065 | 49 | 57 | 0.96 (0.94–0.97) | 0.66 (0.58–0.73) | 0.95 (0.93–0.96) | 0.69 (0.62–0.76) | 0.04 | 0.32 | 91.73 |
| SXT | 912 | 513 | 367 | 7 | 25 | 0.98 (0.96–0.99) | 0.95 (0.93–0.97) | 0.94 (0.91–0.96) | 0.99 (0.97–0.99) | 0.02 | 0.05 | 96.49 |
| TCY | 44 | 40 | 4 | 0 | 0 | 1.00 (0.40–1.00) | 1.00 (0.91–1.00) | 1.00 (0.40–1.00) | 1.00 (0.91–1.00) | 0 | 0 | 100 |
| AZM | 156 | 144 | 12 | 0 | 0 | 1.00 (0.74–1.00) | 1.00 (0.97–1.00) | 1.00 (0.74–1.00) | 1.00 (0.97–1.00) | 0 | 0 | 100 |
| CST | 41 | 41 | 0 | 0 | 0 | - | 1.00 (0.91–1.00) | - | 1.00 (0.91–1.00) | - | 0 | 100 |
| MEM | 132 | 132 | 0 | 0 | 0 | - | 1.00 (0.97–1.00) | - | 1.00 (0.97–1.00) | - | 0 | 100 |

The total number of comparisons, true negatives (TN), true positives (TP), false negatives (FN), false positives (FP), sensitivity, specificity, positive predictive value (PPV), negative predictive value (NPV), very major error (VME) rate, major error (ME) rate, and concordance are shown for ampicillin (AMP), chloramphenicol (CHL), broad-spectrum cephalosporins (CEP), ciprofloxacin (CIP), sulfamethoxazole-trimethoprim (SXT), tetracycline (TCY), azithromycin (AZM), colistin (CST) and meropenem (MEM). Confidence intervals (95%) are shown in parenthesis.

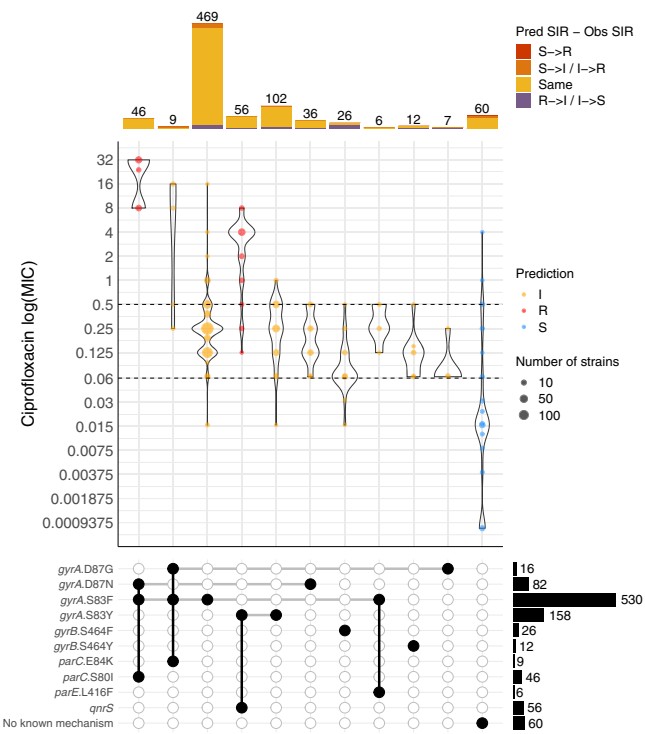

**Fig. 3 Genotypic predictions of antimicrobial resistance.** Distribution of minimum inhibitory concentration (MIC) values (mg L$^{-1}$) for ciprofloxacin in a collection of *S.* Typhi isolates with different combinations of genetic mechanisms that are known to confer resistance to this antibiotic. Only combinations observed in at least five genomes are shown. Dashed horizontal lines on the violin plots mark the CLSI clinical breakpoint for ciprofloxacin. Point colors inside violins represent the genotypic AMR prediction by Pathogenwatch on each combination of mechanisms. Barplots on the top show the abundance of genomes with each combination of mechanisms. Bar colors represent the differences between the predicted and the observed SIR (e.g., red for a predicted susceptible mechanism when the observed phenotype is resistant). S susceptible, I intermediate, and R resistant.

to ciprofloxacin were found in the public genomes but had no or little associated MIC data, including seven additional triple mutations (Supplementary Table 3 and Supplementary Fig. 6).

The user can overlay the AMR predictions on the tree and the map views for one or multiple antibiotics, genes, or SNPs, thus intuitively linking resistance with genome clustering and geographic location. For example, the distribution of genomic predictions of ciprofloxacin-resistant, MDR, or extremely drug resistant (XDR) *S.* Typhi on the map and on the tree of 4389 public genomes highlights the lineages that represent a particular challenge to treatment and their geographical distribution (Supplementary Fig. 7).

MDR and XDR phenotypes have been associated with the acquisition of plasmids in *S.* Typhi[3,20]. Pathogenwatch identifies plasmid replicon sequences in the user genomes and reports them on the genome report and on the typing table in the collection view (Fig. 1). Pathogenwatch reported between one and four plasmid replicon marker sequences in a third of the public genomes (1571/4389, 35.79%, Supplementary Fig. 8a). The cryptic plasmid pHCM2, which does not carry resistance genes[29], was the most common replicon detected amongst genomes in which acquired resistance genes were not detected. The distribution of replicon genes showed that the combination of IncH1A and IncH1B(R27) was prevalent in MDR genomes from Southeast Asia and East Africa belonging to clade 4.3.1, while the same

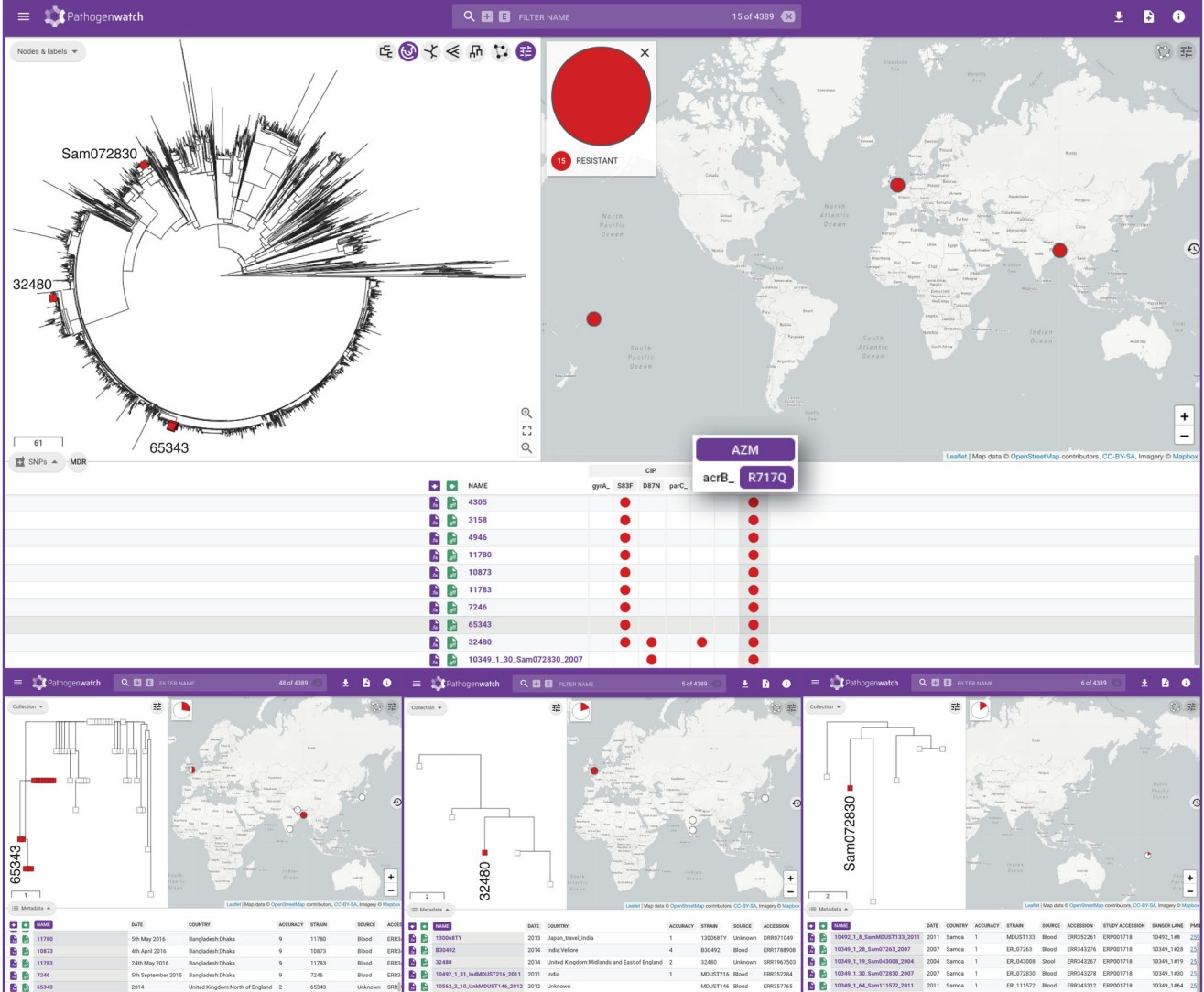

**Fig. 4 Pathogenwatch data reusability.** Fifteen genomes carrying the *acrB*_R717Q mutation recently linked to azithromycin resistance in *S*. Typhi are shown in red on the tree of 4389 public genomes and on the map. The presence of the mutation is also indicated by the red circles on the SNPs table. Three of these genomes (tree labels) belong to isolates collected before the mutation was first described and are shown in more detail in the bottom panels. The data are available at https://pathogen.watch/collection/07lsscrbhu2x-public-genomes.

combination with the addition of IncFIA(HI1) was more prevalent in West Africa, and associated with clade 3.1 (Supplementary Fig. 8b–d). The IncH1A and IncH1B(R27) sequences detect fragments of the *repA2* and *repA* genes, respectively, of the IncHI1 conjugative plasmid which has historically been associated with the majority of MDR typhoid[3]. IncFIA(HI1) detects fragments of the *repE* gene that is present in a subset of IncHI1 plasmids, including the plasmid sequence type PST2 variant common in *S*. Typhi 3.1 in West Africa, but lacking from the PST6 variant that is widespread in *S*. Typhi 4.3.1 in East Africa and Asia[30].

**Maximizing the utility of genomic data.** Azithromycin is one of the last oral treatment options for typhoid for which resistance is currently uncommon, of particular importance in endemic areas with high rates of fluoroquinolone-resistance and outbreaks of XDR *S*. Typhi. A non-synonymous point mutation in the gene encoding the efflux pump AcrB (R717Q) was recently singled out as a molecular mechanism of resistance to azithromycin in *S*. Typhi[31]. Pathogenwatch detected the *acrB*_R717Q mutation in a

collection of 12 Bangladeshi genomes of genotype 4.3.1.1 isolated between 2013 and 2016 in which this mutation was first described (Fig. 4). Notably, Pathogenwatch also detected the *acrB*_R717Q mutation in three additional genomes, two from isolates recovered in England in 2014 (no travel history available[32]), and one from an isolate recovered in Samoa in 2007[3]. The Samoan genome 10349_1_30_Sam072830_2007 was typed as genotype 3.5.4, while the English genomes 65343 and 32480 (no travel information available) belonged to genotypes 4.3.1.1 and 4.3.2.1, respectively. Genome 65343 was closely related to the cluster of 12 genomes from Bangladesh where this mutation was first described, while genome 32480 belonged to a small cluster of five genomes from India or with travel history to India. Thus, reanalysis of public data with Pathogenwatch showed that the *acrB*_R717Q mutation has emerged in multiple genetic backgrounds, in multiple locations, and as early as 2007.

**Pathogenwatch applied to rapid risk assessment.** Typhoid fever is rare in countries with a good infrastructure for the provision of clean water and sanitation, with most cases arising from travel to

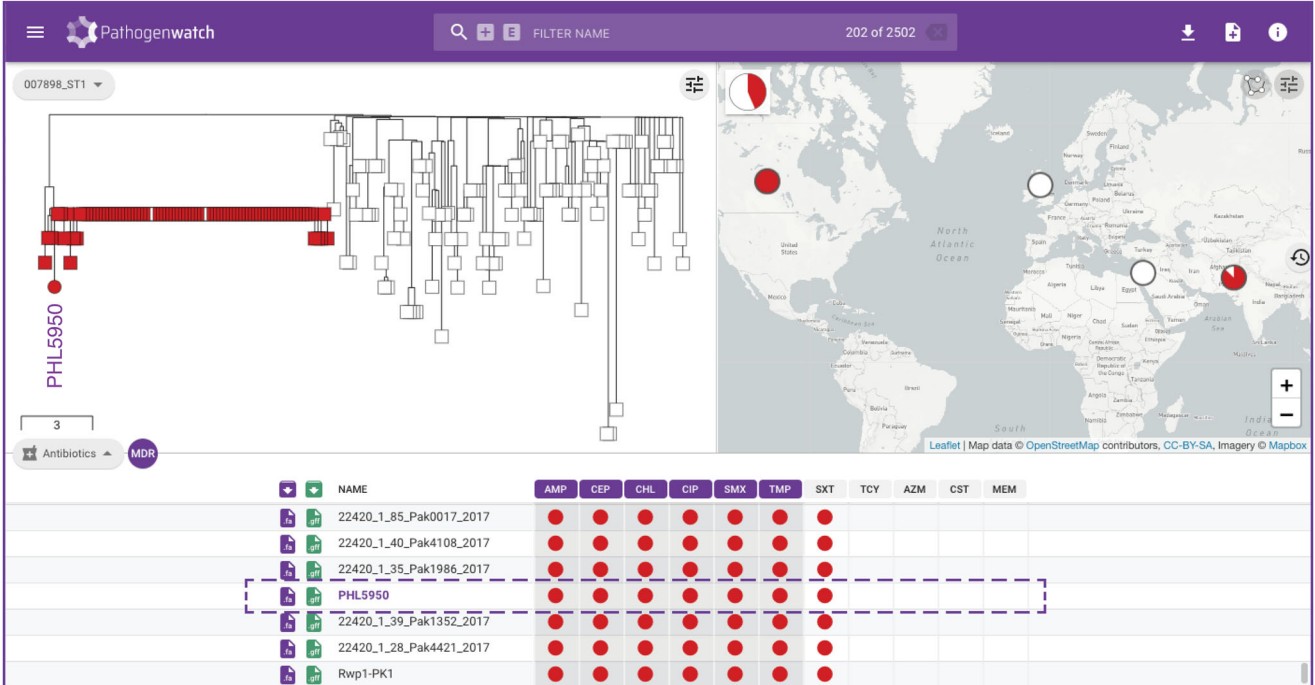

**Fig. 5 Rapid risk assessment of typhoid fever cases in non-endemic regions.** Pathogenwatch places genome PHL5950 from an isolate recovered in Canada and with travel history to Pakistan within the XDR-outbreak in Pakistan. Red markers on the tree and table indicate XDR isolates. The data are available at https://pathogen.watch/collection/11lsok8nrzts-wong-et-al-2018-idcases-15e00492.

endemic areas[33]. Ceftriaxone-resistant typhoid fever was recently reported in developed countries from patients with travel history to Pakistan[34–36]. The isolates were associated to the recent outbreak of XDR *S.* Typhi in the Sindh province of Pakistan by the epidemiological data, the antibiograms, and information derived from WGS of the clinical isolate, such as presence of resistance genes and mobile genetic elements. In some cases, the genomes were contextualized with retrospective genomes by building a phylogenetic tree using an existing bioinformatic pipeline[34,35].

We exemplify how Pathogenwatch facilitates this analysis with the genome from an isolate recovered in Canada (PHL5950, accession RHPM00000000 [https://www.ncbi.nlm.nih.gov/nuccore/RHPM00000000.1/][36]). Pathogenwatch provides a printable genome report (Supplementary Fig. 9) including genotyping and in silico serotyping information, predicted resistance profile, and the presence of resistance genes and mutations. In addition, Pathogenwatch places the genome within the Pakistani XDR outbreak (Fig. 5) and shows the close genetic relatedness (between three and eight pairwise differences) of the isolates via the downloadable score matrix.

**Pathogenwatch as a tool for international collaboration in typhoid surveillance.** As WGS capacity becomes established in typhoid endemic countries, there is a growing opportunity for local genomic surveillance and for collaboration across borders. This is underscored by the growing number of genomes from the Indian Subcontinent (Supplementary Fig. 3), where epidemic clone 4.3.1 (H58) and the nested clade of fluoroquinolone-resistant triple mutants belonging to genotype 4.3.1.2 (H58 lineage II) have been shown to have originated[3,27]. The triple mutants were first reported in Nepal (isolated in 2013–2014) and linked to isolates from India from 2008 to 2012[27] and are still circulating in the region[24,37]. The public data integrated in Pathogenwatch showed that, at the time of writing, this lineage is represented by 195 public genomes from seven countries (India, Bangladesh, Nepal, Pakistan, Myanmar, Japan, and United Kingdom,

Fig. 6a[3,22,26,32,37–40]) and from as early as 2006 (Japan, with travel history to India, Fig. 6b[38]). Linking the tree and the map highlights distinct clusters of genomes that show evidence of transmission across borders, for example between India–Pakistan and India–Nepal (Fig. 6c, d). In addition, Pathogenwatch confirmed the presence of resistance genes *dfrA15*, *sul1*, and *tetA*(A) and the IncN replicon in three genomes from the United Kingdom (two with travel history to India)[26] and, additionally, in two related genomes from Japan with travel history to Nepal and India (Fig. 6b). Altogether, these observations suggest that this lineage circulating in South Asia and linked to treatment failure with fluoroquinolones can acquire plasmids with additional AMR genes, with the concomitant risk of the clonal expansion of a lineage that poses additional challenges to treatment.

## Discussion

Our understanding of the *S.* Typhi population structure, including MDR organisms has improved dramatically since the introduction of WGS providing a much needed level of discrimination for a human-adapted pathogen that exhibits very limited genetic variability. Progress towards the widespread implementation of WGS for epidemiological investigations and integrated routine surveillance within public health settings needs to be accompanied by i) surveillance programs in endemic regions; ii) implementation of WGS at laboratories in endemic regions; iii) analysis of WGS data with fast, robust, and scalable tools that deliver information for public health action; iv) dissemination of WGS data through networks of collaborating reference laboratories at national, international and global scales; and v) provision of WGS data and associated metadata through continuously growing databases that are amenable to interaction and interpretation[41]. Here, we introduced Typhi Pathogenwatch, a web application for genomic surveillance and epidemiology of *S.* Typhi, which enhances the utility of public WGS data and associated metadata by integration into an interactive resource that users can browse or query with their own genomes.

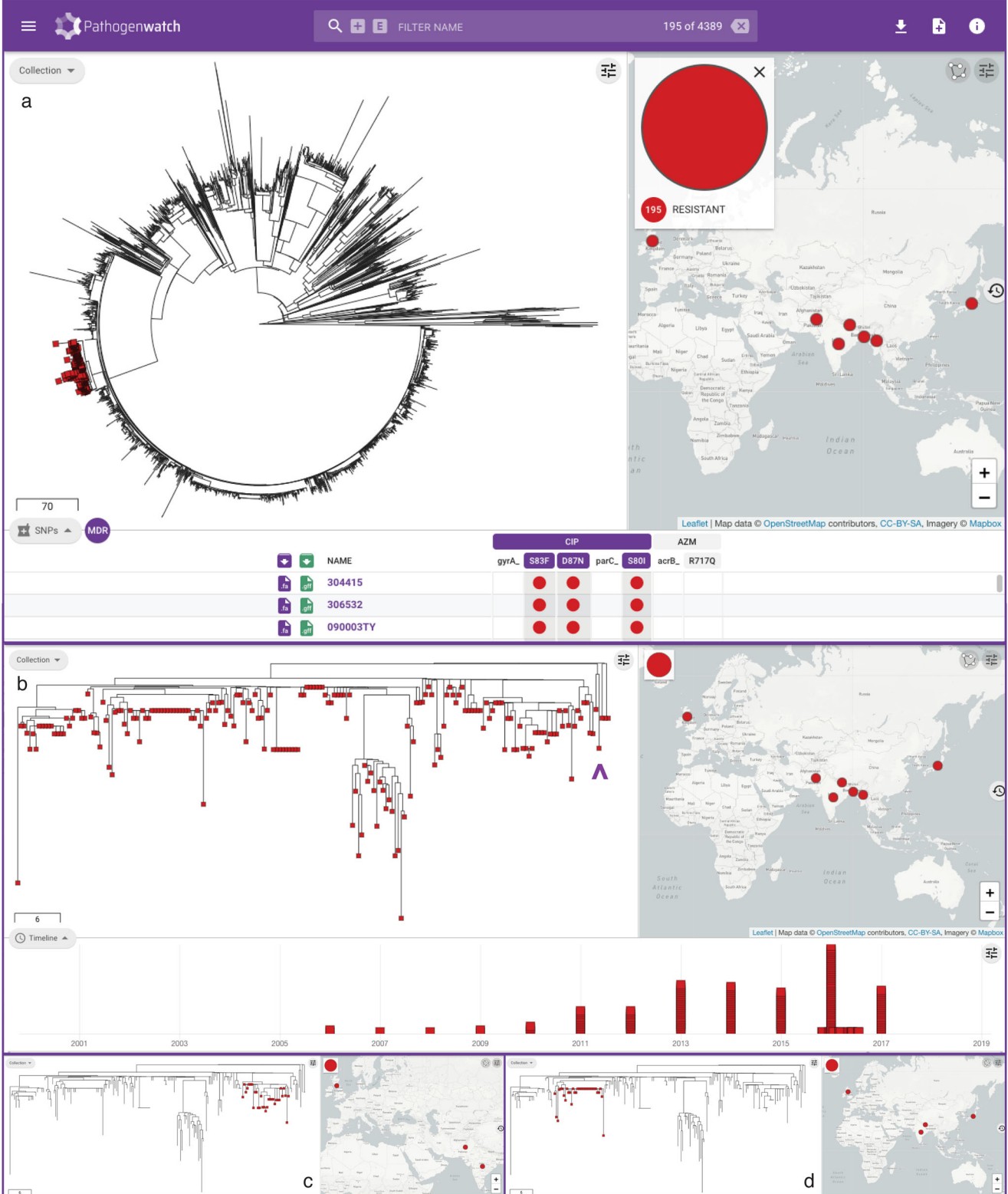

**Fig. 6 Pathogenwatch to for collaborative international surveillance of *S*. Typhi. a** Pathogenwatch highlights 195 ciprofloxacin-resistant triple mutants on the public data tree and map by simultaneously selecting the mutations *gyrA*_S83F, *gyrA*_D87N, and *parC*_S80I on the SNPs table (red markers). **b** Detailed visualization of the triple mutants showing the temporal distribution of the genomes on the timeline. Purple arrowhead: four genomes with *sul1*, *dfrA15*, *tetA* (A) and the IncN replicon from the UK and Japan. Selecting individual clades on the tree shows distinct clades that span neighboring countries India-Pakistan (**c**) and India-Nepal (**d**). The data are available at https://pathogen.watch/collection/07lsscrbhu2x-public-genomes.

We demonstrated that genomic predictions of AMR in Pathogenwatch were highly concordant with the resistance phenotype. A previous study of 332 *S.* Typhi isolates analysed in a single reference laboratory reported only 0.03% discordant results[28] versus 3.66% from our data. Similarly, AMRFinder[7] and Resfinder 4.0[6] reported ≥98.0% overall concordance, but for two large collections of non-typhoidal *Salmonella* genomes. A limitation of our study is that it amalgamated published susceptibility data from thirteen different publications conducted in eight different countries. The availability of consistent laboratory antimicrobial susceptibility testing data is key for the periodic benchmarking and refinement of genomic predictions of AMR[42], as made evident by the different mechanisms and combinations thereof identified for ciprofloxacin. Phenotypic resistance data consistently collected and reported could also be included in the Pathogenwatch metadata table. The unique combination of phenotypic and genotypic resistance with location, time, and population structure could aid the investigations of emerging resistance and discovery of novel resistance mechanisms.

The growing collection of public genomes is updated each time that a novel AMR mechanism is added to the curated Pathogenwatch AMR library. This can potentially reveal the presence of a newly identified gene or mutation in historic isolates, thus maximizing data reusability from which new insights into novel AMR mechanisms can be derived. The utility of maintaining a regularly updated archive of WGS data that can be rapidly "mined" for the presence of newly discovered AMR determinants was elegantly illustrated before by the retrospective discovery of the colistin resistance gene *mcr-1* in *S. enterica* and *Escherichia coli* genomes from Public Health England[43]. With Pathogenwatch the entire Typhi community can access the updated AMR predictions, thus democratizing the reusability of the genomic data.

Contextualizing new genomes with existing data has become a routine part of genomic epidemiology, as it can complement epidemiological investigations to place the new genomes in or out of an outbreak, link to past outbreaks, and determine if the success of a resistant phenotype is the result of a single clonal expansion or multiple independent introductions[44]. Analyzing new genomes in the context of global genomes involves the retrieval, storage, and bioinformatic analysis of large amounts of sequence data and linked metadata, which is time-consuming and largely unfeasible for hospitals or public-health agencies with limited computing infrastructure. We demonstrated how Pathogenwatch circumvents this obstacle using the public genomes to exemplify outbreak investigations in endemic areas and patient management in non-endemic countries with travel history to endemic areas.

The interpretation of the genomic context relies heavily on the completeness of the public collection used for contextualization and of its metadata. The International Typhoid Consortium collected and sequenced around 40% of the global genomes available in Pathogenwatch for comparison[3,4], but local, national, and international genomic surveillance programs are needed for the real-time management of emerging lineages that pose a direct threat to human health[45]. Pathogenwatch does not currently support automated updates or submissions, which instead requires retrieval and curation of genome data and associated metadata. For example, as of November 2020 Pathogenwatch comprises 4234 of 4389 (96.5%) *S.* Typhi genomes with at least both year and country of isolation, while the same applies to 3473 of 7743 (44.9%) genomes on Enterobase[12], 3936 of 5618 (70.1%) genomes on GenomeTrakr (14), and 2085 of 3100 (67.3%) genomes on PATRIC[9]. Pathogenwatch also displays patient travel information when available. While automated updates are needed to ensure the most up-to-date collection of genomes, the provision of genomes with available metadata maximizes the value that

can be derived from the genomes. The metadata linked to the public genomes in Pathogenwatch can be expanded and retrospectively updated following recommendations of the expert community, and buy-in from international surveillance networks to make the metadata available.

Pathogenwatch can facilitate collaborative surveillance in endemic areas via data integration and shared collections for the early detection and containment of high-risk clones. Collections can be set to off-line mode to work while disconnected from the internet, which may be advantageous in settings with unreliable internet connections. Despite recent efforts to promote data openness[46,47], several challenges to sharing genomic data and linked metadata remain in both the academic and public-health settings[41]. User-uploaded genomes, their metadata, and derived collections remain private in the Pathogenwatch user account, unless the user specifically shares them via a collection URL. Users can also integrate private and potentially confidential metadata into the display without uploading it to the Pathogenwatch servers. This private metadata will not be shared even if the collection is set to be shared via web link[48].

Recent improvements in our understanding of the disease burden and the dissemination of AMR in *S.* Typhi, and the development of new typhoid conjugate vaccines have bolstered efforts to employ routine vaccination for the containment of typhoid fever[49]. Routine surveillance coupled with WGS can inform decisions on suitable settings for the introduction of vaccination programs and on the evolution of pathogens in response to them[50,51]. Pathogenwatch should be linked to routine genomic surveillance around typhoid vaccination initiatives to monitor the population dynamics in response to the deployment of new vaccines. The consistent provision of patient demographic data in the metadata would be of particular utility in this context.

Rapid, timely access to information on local patterns of AMR may inform treatment regimens, which could ultimately lead to a reduction in morbidity and mortality associated with enteric fever[52]. Typhi Pathogenwatch combines accurate genomic predictions of AMR with broad geographic and population context within an easy-to-use interface delivered for the community and accessible to users of all bioinformatics skills levels to support ongoing typhoid surveillance programs. The modular architecture of Pathogenwatch allows new functionalities to be added to cater to the community needs.

## Methods

**The Pathogenwatch application**. The Pathogenwatch user interface is a React[53] single-page application with styling based on Material Design Lite v1.3.0[54]. Phylocanvas[55] is used for phylogenetic trees, Leaflet v1.4.0[56] is used for maps, and Sigma v1.2.1[57] is used for networks. The Pathogenwatch back-end, written in Node.js, consists of an API service for the user interface and four "Runner" services to perform analysis: species prediction, single-genome analyses, tree-building, and core genome multilocus sequence typing (cgMLST) clustering. Runner services spawn Docker containers for queued tasks, streaming a FASTA file or prior analysis through standard input and storing JSON data from standard output. Data storage and task queuing/synchronization are performed by a MongoDB cluster.

**S. Typhi genome assemblies**. Genome assemblies can be uploaded by the user in FASTA format or assembled de novo from high-throughput short read data with the Pathogenwatch pipeline[58], as described in the Pathogenwatch documentation[59].
Genomes from published studies with geographical localization metadata and short read data on the European Nucleotide Archive (ENA) are available as public data and accessible to all users for browsing and for contextualization of their own datasets. As of November 2020, 4389 public *S.* Typhi genomes from 26 studies were available (Supplementary Table 1). Genomes were assembled de novo with a previously described assembly pipeline[60]. Briefly, FASTQ files were used to create multiple assemblies using VelvetOptimiser v2.2.5 and Velvet v1.2[61] and/or SPAdes v3.9.0[58] and a range of *k*-mer sizes of 66–90% of the read length (in increments of 4). An assembly improvement step was applied to the assembly with the best N50 and contigs were scaffolded using SSPACE v2.0 and sequence gaps filled using GapFiller v1.11. Assemblies were evaluated based on their metrics and the Pathogenwatch core genome stats (number of contigs, assembly length, N50,

non-ATCG characters, GC content, number of core matches). Seventeen public and published genomes were excluded as the assemblies either contained more than 700 contigs, more than 50,000 non-ATCG characters, a GC content below the smallest GC content or above than the largest GC content of the *S. enterica* subsp *enterica* genomes in RefSeq, or a total length that is <10% smaller than the smallest genome or >10% larger than the largest *S. enterica* subsp *enterica* genome in RefSeq, For five isolates, we used genome assemblies deposited in GenBank that met the same quality criteria. The metadata and assembly stats and method of the public genomes is available on (Supplementary Data 1).

**Pathogenwatch typing of S. Typhi genomes.** For both user-uploaded and public genomes, Pathogenwatch outputs a taxonomy assignment, a map of their locations, and assembly quality metrics. The taxonomy assignment is the best match to a microbial version of the RefSeq genome database release 78, as computed with Mash v2.1[62] ($k = 21$, $s = 400$)[63].

Pathogenwatch also provides compatibility with *Salmonella* serotyping (SISTR[11]), multi-locus sequence typing (MLST[13]), core-genome MLST (cgMLST[8]) and *S.* Typhi single-nucleotide polymorphism (SNP)-based genotyping (GenoTyphi[4]), as detailed in the documentation[64]. The MLST and cgMLST schemes are periodically downloaded from Enterobase[65,66], and samples are typed as described in the documentation[67,68]. Exact allele matches are reported using their allele ID. Multiple allele hits for a gene are reported if present. Inexact allele matches and novel STs are reported by hashing the matching allele sequence and the gene IDs, respectively.

Pathogenwatch implements SISTR (*Salmonella* In Silico Typing Resource[11]), which produces serovar predictions from WGS assemblies by determination of antigen gene and cgMLST gene alleles using blastn v2.2.31+. Pathogenwatch uses the cgmlst_subspecies and serovar fields from the SISTR JSON output to specify the serovar.

Pathogenwatch uses an implementation of GenoTyphi[4,24] designed to work with assemblies to assign *S.* Typhi genomes to a regularly updated predefined set of clades and subclades based on a curated set of SNPs. The blastn v2.2.30 program is used to align the query loci and identify positions of diagnostic SNPs, which are then processed according to the rules of the GenoTyphi scheme[69]. The genotype assignment and the number of diagnostic SNPs identified on the assemblies are reported.

The plasmid replicon marker sequences are detected in the user and public genome assemblies with Inctyper, which uses the PlasmidFinder Enterobacteriaceae database[15], as detailed in the documentation[70].

**The Pathogenwatch S. Typhi core genome library.** Pathogenwatch supports SNP-based neighbor joining trees of *S.* Typhi both for user genomes (collection trees) and public genomes (population tree and subtrees). The trees are inferred using a curated core gene library of 3284 *S.* Typhi genes[71] generated from a pan-genome analysis of 26 complete or high-quality draft genomes (Supplementary Table 4) with Roary v3.2.0[72] and identity threshold of 95%. The core gene families were realigned using MAFFT v7.2.2.0[73], and filtered or trimmed according to the quality of the alignments. The gene with the fewest average pairwise SNP differences to the other family members was selected as the representative for each family. We then selected 19 reference genomes (Supplementary Table 4) belonging to different genotypes according to the population structure previously described[4]. The gene families were searched against each of the 19 reference genomes and filtered according to the following rules: a) only universal families with complete coverage of the representative were kept; b) all paralogues were removed; c) overlapping gene families were merged into a single, contiguous pseudo-sequence. A BLAST[19] core library was then built with the representative genes, and a profile of variant sites determined for the core genes present in each reference genome. Each of the 4389 public genomes was then clustered with its closest reference genome based on this profile of variant sites, thus constituting each of the 19 population subtrees that Pathogenwatch employs to contextualize user-uploaded genomes.

**Pathogenwatch genome clustering of S. Typhi.** The relationships between genomes are represented with trees (dendrograms) based on the genetic distance computed from substitution mutations in the core gene library, as described in detail in the documentation[74]. User-provided assemblies are queried against the *S.* Typhi core gene library with blastn v2.2.30[19] using an identity threshold of 90%. The core gene set of each query assembly is compared to the reference genome core that has the most variant sites in common. An overall relative substitution rate is determined, and loci that contain more variants than expected assuming a Poisson distribution are filtered out. Pairwise distances between assemblies (including user-provided and reference) are scored via a distance scoring algorithm that compares all variant positions from all pairs of core gene sets, SNPs are counted (generating a downloadable pairwise difference matrix) and normalized by the relative proportion of the core present (generating a downloadable pairwise score matrix). The pairwise score matrix is then used to infer a midpoint-rooted neighbor-joining tree using the Phangorn v2.4.0[75] and Ape v5.1[76] R packages. Trees are computed for the user assemblies only (collection tree), and for the user assemblies and public

assemblies assigned to the same reference genome (public data subtrees), all of which are downloadable in Newick format.

We benchmarked the Pathogenwatch clustering method against other methods of SNP-based tree inference with three subsets of published genomes: Dataset I) 118 genomes spanning the population diversity of *S.* Typhi defined by GenoTyphi (Supplementary Data 2); Dataset II) 138 closely related genomes, from a recent clonal expansion of the multidrug-resistant haplotype H58 within Africa (Supplementary Data 3); and Dataset III) 43 strains from clade 3.2 including CT18, the first completed *S.* Typhi genome, which remains reference of choice for most population genomics studies (Supplementary Data 4). For each subset a tree was generated with four different methods: 1) Pathogenwatch; 2) maximum likelihood (ML) with RAxML v8.2.8[77] on SNPs extracted from an alignment of concatenated core genes generated using Roary v3.6.0[72]; 3) neighbor joining (NJ) with FastTree v2.1.8[78] using the option –noml on the same alignment as 2); and 4) ML with RAxML v8.2.8 on SNPs extracted from a previously published CT18-guided alignment[3]. Five hundred bootstrap replicates were computed for the ML trees (methods 2 and 4). We compared the topology of the trees thus generated using the treescape function from the Treescape v1.10.18 R package (now available as Treespace[79]) with the Kendall-Colijn distance and lambda parameter set to 0. The topology of the Pathogenwatch tree from dataset III was compared to the tree from method 4 using the Tanglegram algorithm of Dendroscope v3.5[80]. The tree files used in the tree comparisons are provided in the ref. [81].

Genomes can also be clustered in Typhi Pathogenwatch based on their cgMLST profile using single linkage clustering. Distance scores are calculated between each pair of samples by identifying the genes which have been found in both samples and by counting the number of differences in the alleles. The SLINK algorithm[82] is used to quickly group genomes into clusters at a given threshold. For a given genome, users are able to see how many other genomes it is clustered with at a range of distance thresholds, view the structure of the cluster as a network graph, and view the metadata and analysis for sequences in that cluster.

**Genomic predictions of antimicrobial resistance.** The Pathogenwatch AMR prediction module queries the genome assemblies with blastn v2.2.30[19] for the presence of genes and single point mutations known to confer resistance in *S.* Typhi to ampicillin (AMP), chloramphenicol (CHL), broad-spectrum cephalosporins (CEP), ciprofloxacin (CIP), sulfamethoxazole (SMX), trimethoprim (TMP), the combination antibiotic co-trimoxazole (sulfamethoxazole-trimethoprim, SXT), tetracycline (TCY), azithromycin (AZM), colistin (CST), and meropenem (MEM) (Supplementary Table 2[83]), as detailed in the documentation[84].

The Pathogenwatch AMR prediction module also provides a prediction of AMR phenotype inferred from the combination of identified mechanisms. To benchmark the genotypic resistance predictions, we used a set of 1316 genomes from 16 published studies (Supplementary Table 1) with drug susceptibility interpretation available for at least one of the 12 antibiotics reported by Typhi Pathogenwatch, grouping the Resistant and Intermediate classifications as insusceptible. For each antibiotic, the sensitivity, specificity, positive predictive value (PPV) and negative predictive value (NPV) for detection of known resistance determinants, and their 95% confidence intervals (CI) were calculated with the epi.tests function of the epiR v1.0-14 package[85]. False negative (FN) and false positive (FP) results were further investigated with alternative methods by querying the genome assemblies with Resfinder v3.2.1[25] and/or by mapping and local assembly of the sequence reads to the Bacterial Antimicrobial Resistance Reference Gene Database (Bioproject PRJNA313047) with ARIBA v2.14.4[23].

Seven studies reported ciprofloxacin MICs for a total of 889 *S.* Typhi strains (Supplementary Table 1). We compared the Typhi Pathogenwatch ciprofloxacin resistance predictions for the different combinations of genetic AMR determinants against the MIC values re-interpreted with the ciprofloxacin breakpoints for *Salmonella* spp. from CLSI M100 30th edition (susceptible MIC ≤ 0.06 mg L$^{-1}$; intermediate MIC = 0.12 to 0.5 mg L$^{-1}$; resistant MIC ≥1 mg L$^{-1}$[86]) with a script that is available at ref. [81].

**Reporting summary.** Further information on research design is available in the Nature Research Reporting Summary linked to this article.

## Data availability
The genome assemblies and linked metadata analysed in this study are available from: https://pathogen.watch/collection/07lsscrbhu2x-public-genomes, https://pathogen.watch/collection/g5pbucot6e58-hendriksen-et-al-2015, and https://pathogen.watch/collection/11lsok8nrzts-wong-et-al-2018-idcases-15e00492. The raw sequence data is available from the European Nucleotide Archive via the accessions provided in Supplementary Data 1, and also found in the metadata table of https://pathogen.watch/collection/07lsscrbhu2x-public-genomes.

## Code availability
The tree comparison and AMR benchmarking input files and script are available from https://gitlab.com/cgps/pathogenwatch/publications/-/tree/master/styphi. The Pathogenwatch web application is available at https://pathogen.watch/ and works best on Chromium-based web browsers.

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

## Acknowledgements

We are grateful to Flora Stevens and Joanne Freedman from the Travel Health and IHR department at Public Health England for providing some of the travel information linked to isolates from the United Kingdom, and to Dr. Koji Yahara, Dr. Makoto Ohnishi and Dr. Masatomo Morita for providing the travel information linked to isolates from Japan. Pathogenwatch is developed with support from Li Ka Shing Foundation (Big Data Institute, University of Oxford) and Wellcome (grant number 099202). S.A. and D.M.A. are supported by the National Institute for Health Research (UK) Global Health Research Unit on genomic Surveillance of AMR (16_136_111) and by the Centre for Genomic Pathogen Surveillance (http://pathogensurveillance.net). Z.A.D. received funding from the European Union's Horizon 2020 research and innovation programme under the Marie Skłodowska-Curie grant agreement TyphiNET No 845681. L.S.B. is funded by Plan GenT (CDEI-06/20-B), Conselleria de Sanitat Universal i Salut Pública, Generalitat Valenciana (Valencia, Spain).

## Author contributions

D.M.A. conceived the Pathogenwatch application. C.Y., R.J.G., K.A., B.T., A.U., and D.M.A. developed the Pathogenwatch application. S.A. drafted the manuscript. S.A., D.M.A., K.E.H., S.B., and G.D. contributed to the conception and design of the work, data interpretation, and substantially revised the manuscript. S.A., C.Y., V.K.W., Z.A.D., S.N., A.J.P., J.A.K., S.E.P., and F.M. contributed to the acquisition and interpretation of data. S.A., C.Y., and L.S.B. analysed the data. All authors read and approved the final manuscript.

## Competing interests

The authors declare no competing interests.
