## [Peer Review File · Nature Communications]

REVIEWER COMMENTS

Reviewer #1 (Remarks to the Author):

The manuscript describes an interface that uses the known whole genome sequences for *S. Typhi* and allows a person or public health lab to incorporate their data easily into the public data. This is an important step towards making the basic research data of WGS accessible to labs that do not have significant informatics capabilities. The manuscript is well written, worth publishing public health interfaces for WGS data are important.

However, for the interface to be applicable to the countries that need it most. Right now most isolates are found in countries without easy access to WGS. Thus, it is important for the interface to provide directions for either PCR based or microbiologically based tests that would differentiate between lineages known to be in a region. In addition, the interface should have a decision tree and subsequent tests to confirm the lineage and if there is an unexpected result to ask for the isolate's DNA to be sent to a reference lab where the WGS can be performed at no cost to the sender of the DNA.

Sequential numbering of all pages (as was done for the supplementary material, but not for the main text) would be useful for the reviewer, especially if re-review is needed.

Page 7, line 5 most readers will not know what tree space is or how it is defined. Please define it. Also there should be a measure of tree topology similarity. The figure is nice, but it is supplementary and requires extra effort by the reader to examine it. Supplemental figure 2a & b, the axes need to be labeled.

The cladogram (supp figure 2c) is nice but given the large number of nodes with bootstrap values of 100 it is no surprise that the different methods yield similar trees.

Previous studies have shown that the ends of contigs often contain genotyping errors. Did the authors remove the last kilobase of each contig and all contigs less than 2 kb for their WGS analyses?

Reviewer #2 (Remarks to the Author):

Thank you for the opportunity to review "A global resource for genomic predictions of antimicrobial resistance and surveillance of *Salmonella Typhi* at Pathogenwatch" by Argimón and colleagues. The paper presents Typhi Pathogenwatch, a web interface that automates genomic analysis and visualization of *Salmonella Typhi* in the context of published genomes. The tool has extensive capability (e.g., serotyping, AMR identification, phylogenetic inference, etc), which the authors highlight using a number of vignettes. In addition, they present extensive discussion regarding the applications of Pathogenwatch, especially in areas where there are limited resources (e.g., in locales that may be able to generate WGS data but not the computational resources to analyze it at scale). Overall, I find this manuscript to be concise and easy to read with broad appeal to researchers and public health epi/lab practitioners interested in genomic epidemiology. In addition to reviewing the technical aspects of the manuscript, I also tested the web application with the data collections provided by the authors. I find it intuitive and easy to navigate. I was also able to replicate the results the authors present in the manuscript. In addition to the presentation of the functionality of Pathogenwatch, I appreciated the more detailed technical aspects that benchmark the analysis tools available in the application against existing gold standards in the field. The two most important of these are the genotypic AMR predictions and the phylogenetic clustering, for which the authors thoroughly compare the sensitivity and specificity (for the AMR prediction) and the concordance with other methods (for the phylogenetic inference). Of note, the ability to add data incrementally to the existing datasets is a tremendous advancement from other methods. In general, the manuscript is technically sound, and I only have a few questions/comments regarding parts of the analysis and discussion.

1) A large focus of the manuscript was on the ability to predict antimicrobial resistance (side note, I particularly liked Figure 3). The sensitivity and specificity was based on phenotypic data provided

in a set of published studies. I personally empathize with difficulty in merging results from varied testing approaches and often finding SIR in place of MIC values. Currently, it does not appear that there is a centralized effort to collect phenotypic resistance data in Pathogenwatch, which would limit the ability to identify novel resistance methods. This is addressed in the discussion where the authors state that the AMR database can be easily updated, but it doesn't sound like Pathogenwatch is being tasked with identifying these mechanisms. Can the authors address this more in the discussion? Is it because of data privacy concerns, the need for extensive curation, or variation in phenotypic testing methods that they feel Pathogenwatch isn't able to identify emerging resistance?

2) Along with the comment above, can the authors discuss a minimum dataset that they envision would balance data privacy concerns with broad public health utility? On the website, they recommend including latitude, longitude, year, month, day, but perhaps source, age or age group, and vaccine history would be beneficial for the public health goals of Typhi genomic surveillance. Certainly, it is appealing that users can create fully offline private projects, but when they do decide to publish, what data would they like to see? I could see a couple sentences being added to the discussion on Page 16.

3) The methods for determining the Pathogenwatch S. Typhi core genome library are presented in the methods. Do the authors envision a need to update this periodically as diversity increases among S. Typhi or is it robust enough that it can remain static? Is there a method for determining this threshold?

4) It appears that some of the figures may be from screenshots of the web application. In PathogenWatch, I see that I am able to export images from each panel of the GUI. Is there an option to export the entire view as it is shown in the manuscript? This would be helpful for generating reports that would be distributed in pdf or similar format.

5) In Supplemental Figure 2 the authors present the TreeSpace comparison of the different methods for inferring the phylogeny. Is there are way to statistically test the variation? Certainly from the tanglegram and treespace plots, there is essentially no meaningful difference, but I am personally interested in what test could be performed to assess the significance of the Kendall-Colin distance.

RESPONSE TO REVIEWER COMMENTS

Reviewer #1 (Remarks to the Author):

The manuscript describes an interface that uses the known whole genome sequences for *S. Typhi* and allows a person or public health lab to incorporate their data easily into the public data. This is an important step towards making the basic research data of WGS accessible to labs that do not have significant informatics capabilities. The manuscript is well written, worth publishing public health interfaces for WGS data are important.

However, for the interface to be applicable to the countries that need it most. Right now most isolates are found in countries without easy access to WGS. Thus, it is important for the interface to provide directions for either PCR based or microbiologically based tests that would differentiate between lineages known to be in a region. In addition, the interface should have a decision tree and subsequent tests to confirm the lineage and if there is an unexpected result to ask for the isolate's DNA to be sent to a reference lab where the WGS can be performed at no cost to the sender of the DNA.

We are very grateful to the reviewer for their comments to our manuscript. We agree that information derived from whole-genome sequences can be leveraged to develop diagnostic molecular laboratory tests that would allow researchers and public health staff without access to WGS to identify specific genotypes/lineages. We also appreciate that a web application with a built-in decision tree could aid the interpretation of a PCR-based genotyping test and the identification of novel emerging lineages. The reviewer describes a valuable, yet entirely distinct body of work that is beyond the scope of this manuscript, which describes a web application developed to analyse whole genomes and extract genotyping information, presence of AMR genes and SNPs, and of plasmid sequences, and interpret the data within a broader geographical context. We thank the reviewer for the helpful suggestion and we will discuss it further for inclusion in our development roadmap.

Sequential numbering of all pages (as was done for the supplementary material, but not for the main text) would be useful for the reviewer, especially if re-review is needed.

We have changed the line numbering to continuous across all pages, as requested by the reviewer.

Page 7, line 5 most readers will not know what tree space is or how it is defined. Please define it.

We have added an explanation to this sentence, which is now found in page 7 lines 126-129 and reads:

"In addition, we found that the Typhi Pathogenwatch clustering algorithm produced trees comparable to established methods based on the tree space (visualisations of pairwise distances between trees in two or three dimensions) and the tree topology (Supplementary Figure 2)."

We have also added more information on the Methods section, page 22 lines 521-528. The sentence now reads:

"We compared the topology of the trees thus generated using the treescape function from the Treescape v1.10.18 R package (now available as Treespace⁷⁹) with the Kendall-Colijn distance and $\lambda=0$. The topology of the Pathogenwatch tree from dataset III was compared to the tree from method 4 using the Tanglegram algorithm of Dendroscope v3.5⁸⁰."

Also there should be a measure of tree topology similarity. The figure is nice, but it is supplementary and requires extra effort by the reader to examine it.

The Kendall-Colijn distance used to compare the trees is indeed a measure of tree topology similarity (for more details see <https://onlinelibrary.wiley.com/doi/epdf/10.1111/1755-0998.12676>) and the multidimensional scaling (MDS) plots based on this measure (Supplementary Figure 2) intuitively show that the Pathogenwatch trees are found within the same tree space occupied by

trees generated with gold-standard methods. Nevertheless, we have added a matrix with the pairwise Kendall-Colijn distances between Pathogenwatch and the best trees from the reference methods to Supplementary Figure 2 for the benefit of the readers who would prefer to see the values.

Supplemental figure 2a & b, the axes need to be labeled.

The axes of the MDS plots were labelled x, y, and z in panel **a**, and Axis 1 and Axis 2 in panel **b**. We changed the labels in panel **b** to x and y for consistency. We have also modified the legend to specify that panels **a** and **b** are MDS plots, which should make it clear that the labels correspond to space dimensions. The legend now reads:

“Supplementary Figure 2. a Three-dimensional multidimensional scaling (MDS) plot of pairwise tree distances from dataset I (118 genomes), and b Two-dimensional MDS plot of pairwise tree distances from dataset II (138 genomes), both generated with Treescap.”

The cladogram (supp figure 2c) is nice but given the large number of nodes with bootstrap values of 100 it is no surprise the the different methods yield similar trees.

Previous studies have shown that the ends of contigs often contain genotyping errors. Did the authors remove the last kilobase of each contig and all contigs less than 2 kb for there WGS analyses?

Although we have not filtered the assemblies as per the thresholds described by the reviewer, Pathogenwatch does not utilize the entire assembled sequence for detection of variable positions and tree inference, but based on a predetermined set of 3284 core genes. These may be located in different regions of the assembly contigs in different genomes. However, Pathogenwatch implements a filtering step to remove core gene matches containing an unusual amount of variation due to assembly errors that may prove problematic for tree building. This filtering step accounts for assembly errors caused by a number of variables (such as low sequencing coverage, average read length, and sequencing error rates) and is applied regardless of the position of the core match within the contig. This is mentioned in the Methods section (page 21, lines 474-476) and explained in detail in the Pathogenwatch documentation (<https://cgps.gitbook.io/pathogenwatch/technical-descriptions/core-genome-tree/core-filter>).

Reviewer #2 (Remarks to the Author):

Thank you for the opportunity to review “A global resource for genomic predictions of antimicrobial resistance and surveillance of Salmonella Typhi at Pathogenwatch” by Argimón and colleagues. The paper presents Typhi Pathogenwatch, a web interface that automates genomic analysis and visualization of Salmonella Typhi in the context of published genomes. The tool has extensive capability (e.g., serotyping, AMR identification, phylogenetic inference, etc), which the authors highlight using a number of vignettes. In addition, they present extensive discussion regarding the applications of Pathogenwatch, especially in areas where there are limited resources (e.g., in locales that may be able to generate WGS data but not the computational resources to analyze it at scale). Overall, I find this manuscript to be concise and easy to read with broad appeal to researchers and public health epi/lab practitioners interested in genomic epidemiology. In addition to reviewing the technical aspects of the manuscript, I also tested the web application with the data collections provided by the authors. I find it intuitive and easy to navigate. I was also able to replicate the results the authors present in the manuscript. In addition to the presentation of the functionality of Pathogenwatch, I appreciated the more detailed technical aspects that benchmark the analysis tools available in the application against existing gold standards in the field. The two most important of these are the genotypic AMR predictions and the phylogenetic clustering, for which the authors thoroughly compare the sensitivity and specificity (for the AMR prediction) and the concordance with other methods (for the phylogenetic inference). Of note, the ability to add data incrementally to the existing datasets in a tremendous advancement from other methods. In

general, the manuscript is technically sound, and I only have a few questions/comments regarding parts of the analysis and discussion.

We greatly appreciate the reviewer taking the time to thoroughly evaluate both the manuscript and the application.

1) A large focus of the manuscript was on the ability to predict antimicrobial resistance (side note, I particularly liked Figure 3). The sensitivity and specificity was based on phenotypic data provided in a set of published studies. I personally empathize with difficulty in merging results from varied testing approaches and often finding SIR in place of MIC values. Currently, it does not appear that there is a centralized effort to collect phenotypic resistance data in Pathogenwatch, which would limit the ability to identify novel resistance methods. This is addressed in the discussion where the authors state that the AMR database can be easily updated, but it doesn't sound like Pathogenwatch is being tasked with identifying these mechanisms. Can the authors address this more in the discussion? Is it because of data privacy concerns, the need for extensive curation, or variation in phenotypic testing methods that they feel Pathogenwatch isn't able to identify emerging resistance?

We initially attempted to provide antimicrobial susceptibility data linked to the genomes in Pathogenwatch, but were discouraged by the lack of consistency and harmonization in reporting phenotypic data. We agree with the reviewer that this would be of great utility and, in fact, MIC data is included in Pathogenwatch for other organisms like *Neisseria gonorrhoeae*, where the majority of the isolates sequenced are concentrated in the "global north" and MIC data is consistently reported. However, in recent years multi-country networks for typhoid surveillance have been created, such as SEAP, STRATAA, and SETA (see Carey et al, 2020 DOI: 10.1093/cid/ciaa367) and we anticipate that this will ultimately result in more consistent phenotypic data. The authors of the manuscript are also involved in establishing a Global Typhoid Genomics Consortium whose goals include harmonization of phenotypic resistance data and metadata. The metadata linked to the genomes in Pathogenwatch can easily be updated in the future to reflect this. We have elaborated on this in the Discussion (page 14, lines 314-328).

2) Along with the comment above, can the authors discuss a minimum dataset that they envision would balance data privacy concerns with broad public health utility? On the website, they recommend including latitude, longitude, year, month, day, but perhaps source, age or age group, and vaccine history would be beneficial for the public health goals of Typhi genomic surveillance. Certainly, it is appealing that users can create fully offline private projects, but when they do decide to publish, what data would they like to see? I could see a couple sentences being added to the discussion on Page 16.

We also agree with the reviewer that including patient demographic information would also be of great utility, in particular patient age for vaccination studies. As discussed above, the metadata currently included in Pathogenwatch is largely the result of availability. We included the source (blood, stool, etc.) because that is more consistently reported in the metadata linked to the genomes or it can be derived from the publications through curation. This is an ongoing discussion with experts in the typhoid community and in the future we could incorporate (and retrospectively update) demographic information in Typhi Pathogenwatch. We added two sentences in the Discussion, page 16, lines 369-372, and page 17, line 393-394.

3) The methods for determining the Pathogenwatch *S. Typhi* core genome library are presented in the methods. Do the authors envision a need to update this periodically as diversity increases among *S. Typhi* or is it robust enough that it can remain static? Is there a method for determining this threshold?

The robustness of the core genome library was determined through the benchmarking of the tree building method. *S. Typhi* is a human-adapted pathogen that exhibits limited genetic variability when compared to other species and even other *S. enterica* serovars, but even as genetic diversity increases the core genome will not expand, by definition. Pathogenwatch stats report both the

number and % of core matches found in each genome as well as the % non-core (i.e. accessory genome). If in the future significant deviations were detected in a substantial amount of new genomes, we could reassess the core genome library and update it if needed.

4) It appears that some of the figures may be from screenshots of the web application. In PathogenWatch, I see that I am able to export images from each panel of the GUI. Is there an option to export the entire view as it is shown in the manuscript? This would be helpful for generating reports that would be distributed in pdf or similar format.

We regret that this option is not currently available in Pathogenwatch, but we appreciate the reviewer's comment and we will add it to our development roadmap.

5) In Supplemental Figure 2 the authors present the TreeSpace comparison of the different methods for inferring the phylogeny. Is there are way to statistically test the variation? Certainly from the tanglegram and treespace plots, there is essentially no meaningful difference, but I am personally interested in what test could be performed to assess the significance of the Kendall-ColinJ distance.

Please note that, as per Reviewer #1's request we now provide a matrix pairwise distances between the trees in Supplementary Figure 2. The variation within the tree space could be assessed with the findGroves function of the treespace package, which identifies clusters of trees within the tree space based on the Kendall-Colijn distance --or other distance measures also available. The findGroves function uses hierarchical clustering to identify discrete clusters that may indicate that the data supports several distinct phylogenies. However this analysis was beyond the scope of our benchmarking, which aimed to show that the Pathogenwatch trees are found within the same tree space occupied by trees generated with gold-standard methods.